# Peripheral circadian rhythms in the liver and white adipose tissue of mice are attenuated by constant light and restored by time-restricted feeding

**Daisuke Yamamuro[1], Manabu Takahashi[1], Shuichi Nagashima[1], Tetsuji Wakabayashi[1], Hisataka Yamazaki[1], Akihito Takei[1], Shoko Takei[1], Kent Sakai[1¤a], Ken Ebihara[1], Yusaku Iwasaki[2¤b], Toshihiko Yada[2¤c], Shun Ishibashi[1]** *

**1** Division of Endocrinology and Metabolism, Department of Medicine, Jichi Medical University, Tochigi, Japan, **2** Division of Integrative Physiology, Department of Physiology, Jichi Medical University School of Medicine, Shimotsuke, Tochigi, Japan

¤a Current address: Division of Dermatology and Cutaneous Surgery, Miami Itch Center, University of Miami, Miami, Florida, United States of America
¤b Current address: Laboratory of Animal Science, Graduate School of Life and Environmental Sciences, Kyoto Prefectural University, Sakyo-ku, Kyoto, Japan
¤c Current address: Division of Integrative Physiology, Kansai Electric Power Medical Research Institute, Chuo-ku, Kobe, Japan
* ishibash@jichi.ac.jp

**Data Availability Statement:** All relevant data are within the manuscript and its Supporting Information files.

## Abstract

Disturbance of circadian rhythms underlies various metabolic diseases. Constant light exposure (LL) is known to disrupt both central and peripheral circadian rhythms. Here, we attempted to determine whether the effects of LL are different between various peripheral tissues and whether time-restricted feeding restores the circadian rhythms especially in white adipose tissue (WAT). Six-week-old mice were subjected to three feeding regimes: *ad libitum* feeding under light/dark phase (LD), *ad libitum* feeding under LL cycle, and restricted feeding at night-time under LL cycle with a normal chow. After 3 weeks, we compared body weight, food intake, plasma levels of lipids and glucose, and the expression patterns of the clock genes and the genes involved in lipid metabolism in the liver and WAT. The mice kept under LL with or without time-restricted feeding were 5.2% heavier (p<0.001, n = 16) than the mice kept under LD even though the food intakes of the two groups were the same. Food intake occurred mostly in the dark phase. LL disrupted this pattern, causing disruptions in circadian rhythms of plasma levels of triglycerides (TG) and glucose. Time-restricted feeding partially restored the rhythms. LL eliminated the circadian rhythms of the expression of the clock genes as well as most of the genes involved in lipid metabolism in both liver and WAT. More notably, LL markedly decreased not only the amplitude but also the average levels of the expression of the genes in the liver, but not in the WAT, suggesting that transcription in the liver is sensitive to constant light exposure. Time-restricted feeding restored the circadian rhythms of most of the genes to various degrees in both liver and WAT. In conclusion, LL disrupted the peripheral circadian

**Funding:** This work was supported by a Grant-in-Aid for Scientific Research and MEXT-Supported Program for the Strategic Research Foundation at Private Universities 2011-2015 "Cooperative Basic and Clinical Research on Circadian Medicine" from the Ministry of Education, Culture, Sports, Science and Technology of Japan.

**Competing interests:** There is nothing to disclose.

rhythms more severely in liver than in WAT. Time-restricted feeding restored the circadian rhythms in both tissues.

## Introduction

Circadian rhythm is important for various physiological functions such as sleep, feeding behavior, endocrine function and autonomic nervous activity [1]. In mammals, a central clock in the suprachiasmatic nucleus (SCN) synchronizes peripheral clocks in many organs including the liver, adipose tissue and heart. Circadian rhythm is generated by a cell-autonomous transcriptional-translational feedback loop (TTFL), which is composed of transcriptional factors CLOCK and BMAL1 and nuclear hormone receptors. CLOCK and BMAL1 bind as heterodimers to cis-acting E boxes in the promoters of their own repressors, CRYPTOCHROME (CRY1 and CRY2) and PERIOD (PER1, PER2 and PER3). The nuclear hormone receptors include REV-ERB and the retinoid-related orphan receptor (ROR). PER/CRY heterodimer repress the transcriptional activity of CLOCK/BMAL1 [2]. RORs and REV-ERBs activate and repress the *Bmal1* expression by binding to its RRE element, respectively. This feedback loop permits rhythmic expression of these oscillator genes at various phases with a period of approximately 24 h.

The mammalian circadian clock can be reset by external signals, such as light exposure (light/dark phase) or nutrient signal (feeding time, nutrient status) via a network that operates in both the brain and peripheral tissues. Light exposure is the strongest external signal that synchronizes behavioral and physiological rhythms in mammals. In mice, constant light exposure reduces locomotor activity and the amplitude of behavioral circadian rhythms probably by desynchronizing individual neurons of the SCN [3, 4]. While the oscillator in the SCN is susceptible to light exposure, the oscillator in the liver is less susceptible to light exposure in the short term. Its amplitudes are rather synchronized by external signals generated by feeding [5, 6]. A constant high-fat diet across 24 h light/dark cycles in mice disrupted the rhythm of the expressions of clock genes in the liver [7, 8]. Restricted feeding strongly synchronized the expression of circadian clock genes in the liver without altering the SCN clock function [9]. Although external signals are known to affect the expression of peripheral circadian clock genes, it is not known whether the susceptibility to disruption of the peripheral circadian clock by constant light exposure is different between different organs. It is also unknown whether time-restricted feeding can restore the disrupted circadian rhythms.

Non-alcoholic fatty liver disease (NAFLD), which is characterized by the excessive accumulation of triglycerides (TG) in the liver, is pandemic worldwide affecting about 20~30% of individuals in Western countries [10]. Disruption of circadian rhythms underlie the development of NAFLD. Hepatic TG contents are determined by many factors including supply of free fatty acids (FFAs) from lipolysis in white adipose tissues (WAT), hepatic *de novo* lipogenesis (DNL) and hepatic β-oxidation of FFAs. WAT lipolysis is primarily mediated by adipose triglyceride lipase (ATGL) and hormone-sensitive lipase (LIPE) [11, 12]. On the contrary, lipolysis is counteracted by peroxisome proliferator-activated receptor (PPAR) γ by stimulating adipogenesis [13]. DNL is mediated by lipogenic enzymes such as fatty acid synthase (FASN) and diacylglycerol acyltransferase (DGAT), both of which are transcriptionally regulated by sterol regulatory element binding protein (SREBP)1c and Liver X receptors (LXRs) [14]. On the other hand, many of the genes involved in β-oxidation of FFAs are transcriptionally regulated by PPARα [15], which is a direct target of CLOCK/BMAL1 heterodimer [16]. In addition, liver

actively synthesizes cholesterol in a robust diurnal manner [17] by HMG-CoA reductase (HMGCR) [18], which is transcriptionally regulated by SREBP2 [14].

Here, we aimed to compare the effects of constant light exposure on the peripheral circadian rhythms between liver and WAT. We further aimed to determine whether restricting the time of feeding (as opposed to feeding *ad libitum*) can rescue the peripheral circadian rhythms that have been disrupted by constant light exposure. To this end, we fed mice under three different conditions: *ad libitum* feeding under a light/dark phase (LD cycle), *ad libitum* feeding under a constant light phase (LL cycle), and restricted feeding at night-time under an LL cycle and measured the hepatic expression of the clock genes and genes involved in lipid metabolism.

## Materials and methods

### Animals

Male C57BL/6J mice were obtained from CLEA Japan (Tokyo, Japan). All animal procedures were performed with the approval of the Institutional Animal Care and Research Advisory Committee at Jichi Medical University. All efforts were made to minimize animal suffering.

### Schedule of feeding and light shifts

All mice were kept at controlled temperature (24 ± 1˚C) and humidity (50 ± 10%) with a light 12 h / dark 12 h (light phase 07:00–19:00/dark phase 19:00–07:00 cycle) in a room kept under a condition equivalent to SPF. Light conditions were as follows: Two fluorescent lamps FHF32EX-N-H 32W (Panasonic, Japan) were attached to the ceiling at 2 places in a 6.5 feet × 20 feet room. Light irradiation were 3520 lumens. The ceiling lamps were about 3.3 feet above the cages. In this study we used real time. We did not use zeitgeber time (ZT), because the behavior of the mice might be arrhythmic during the LL cycles. Six-week-old mice were housed individually in a 18 cm x 7 cm x 13 cm tall cage with bedding made of fir chips (Japan SLC, Inc., Hamamatsu, Japan) for 1 week to adapt to the housing conditions and fed a regular diet (CE-2; CLEA Japan, Tokyo, Japan) and water *ad libitum*. After adaptation, the mice were confirmed ostensibly healthy and randomly assigned to the following three conditions of light phase and feeding time for 3 weeks: i) *ad libitum* under LD cycle (light phase 12 h / dark phase 12 h per day), ii) *ad libitum* under LL cycle (constant light phase 24 h per day), iii) time-restricted feeding (access to food between 19:00 to 07:00 under LL cycle (constant light phase 24 h per day). Food intake was measured at 07:00 and 19:00. Sixteen mice were kept under each condition. We started the experiments for each condition at different day in order to minimize the time required for collection of blood and organs.

### Blood and tissue collection

After 3 weeks of treatment, when the mice were 10 weeks old, randomly selected 4 mice were euthanized at each time point 08:00, 14:00, 20:00 and 02:00 the next day by cervical dislocation. We assigned 4 mice to each time point, because this was minimal number for meaningful statistical analyses. Blood, liver and epidydimal fat pads that were used to represent WAT were collected at the time of euthanasia, plunged in liquid nitrogen and stored at -80 ˚C.

### Biochemical analysis of plasma

Blood glucose levels were measured with a free style blood glucose monitoring system (NIPRO, Osaka, Japan). Plasma total cholesterol (TC) was measured using Determiner L TC II kit (Kyowa Medex, Tokyo, Japan). Plasma triglyceride (TG) and non-esterified fatty acid

(NEFA) levels were measured using L-Type TG M kit and NEFA C-test kit (Wako, Osaka, Japan).

## Biochemical analysis of liver and WAT

Frozen liver and WAT were thawed, homogenized in PBS, and mixed with a solution of chloroform/methanol (2:1). The mixture was swirled on a rotating at 4 ˚C for 12 h, and centrifuged at 3,000 x g for 15 min. The lower layer was obtained and evaporated at 40 ˚C to dryness under a stream of nitrogen gas. The residue was dissolved in ethanol, and tissue contents of TC, TG, NEFA and glycerol were measured using Determiner L TC II kit (Kyowa Medex, Tokyo, Japan), L-Type TG M kit (Wako, Osaka, Japan), NEFA C-test kit (Wako, Osaka, Japan) and glycerol colorimetric assay kits (Cayman chemical, Michigan, USA), respectively, following the manufacturer's instructions.

## RNA extraction and quantitative real-time PCR

Total RNA was prepared from mouse liver using TRIzol (Invitrogen). Relative amounts of mRNA were calculated using a standard curve or the comparative cycle threshold method with the Step One Plus Real-Time PCR instrument (Applied Biosystems) according to the manufacturer's protocol. Mouse glyceraldehyde 3-phosphate dehydrogenase (Gapdh) mRNA was used as the invariant control. Although *Gapdh* has been reported to exhibit circadian oscillation in both liver and WAT [19], we did not observe significant oscillations of *Gapdh* in either liver or WAT (data not shown). Primer sequences are shown in Table 1.

**Table 1. Primer sequences used for qRT-PCR.**

| Gene | Forward | Reverse |
|---|---|---|
| *Clock* | 5'-ACTTCCATCTGTCATGATCGC-3' | 5'-ACATAAAGAGACCACTGCACAG-3' |
| *Bmal1* | 5'-CCTTGCATTCTTGATCCTTCC-3' | 5'-TGCCACTGACTACCAAGAAAG-3' |
| *Cry1* | 5'-CTCACTCAAGCAAGGGAGAAG-3' | 5'-CCCACACGCTTTCGTATCA-3' |
| *Per1* | 5'-CTTTGCTTTAGATCGGCAGTG-3' | 5'-CTTCCTCAACCGCTTCAGA-3' |
| *Rorα* | 5'-AGGCACGGCACATCCTAATAA-3' | 5'-ATGGAGCTGTGTCAAAATGATCA-3' |
| *Rorγ* | 5'-CAGATGTTCCACTCTCCTCTTCTCT-3' | 5'-CACGGCCCTGGTTCTCAT-3' |
| *Rev-erbα* | 5'-TTCCCAGATCTCCTGCACAGT-3' | 5'-CAAGGCAACACCAAGAATGTTC-3' |
| *Rev-erbβ* | 5'-ATGGAGACTTGCTCATAGGACACA-3' | 5'-AGTAGGTGGATGTTCTCAGACTGAGA-3' |
| *Hmgcr* | 5'-CGTCATTCATTTCCTCGACAAA-3' | 5'-AGCAGAAAAAAGGGCAAAGCT-3' |
| *Dgat1* | 5'-TCCGCCTCTGGGCATTC-3' | 5'-GAATCGGCCCACAATCCA-3' |
| *Dgat2* | 5'-TGGAACACGCCCAAGAAAG-3' | 5'-CACACGGCCCAGTTTCG-3' |
| *Fasn* | 5'-GCTGCGGAAACTTCAGGAAAT-3' | 5'-AGAGACGTGTCACTCCTGGACTT-3' |
| *Lipe* | 5'-CATATCCGCTCTCCAGTTGACC-3' | 5'-CCTATCTTCTCCATCGACTACTCC-3' |
| *Atgl* | 5'-GGAAGTTGGGTGCCACTTCG-3' | 5'-GGTGCTCTCAGATCTTTGG-3' |
| *Pparα* | 5'-TGCAACTTCTCAATGTAGCCT-3' | 5'-AATGCCTTAGAACTGGATGACA-3' |
| *Pparγ* | 5'-TCTTAACTGCCGGATCCA CAA-3' | 5'-GCCCAAACCTGATGGCA TT CAA-3' |
| *Lxrα* | 5'-GCTCTGCTCATTGCCATCAG-3' | 5'-TGTTGCAGCCTCTCTACTTGGA-3' |
| *Srebp1c* | 5'-TGGATTGCACATTTGAAGACATG-3' | 5'-GGCCCGGGAAGTCACTGT-3' |
| *Srebp2* | 5'-CCGGTCCTCCATCAACGA-3' | 5'-TGGCATCTGTCCCCATGACT-3' |
| *Sp1* | 5'-GCCTCCAGACCATTAACCTC-3' | 5'-CATGAAGACCAAGTTGAGTTCC-3' |
| *Gapdh* | Taqman® Gene Expression Assays (Assay ID: Mm99999915_g | |

## Statistical analyses

Results are presented as the mean ± SD. Since the number of each group (16 or 4) was too small to test whether the values were distributed normally, we employed parametric analyses *a priori*. Statistical differences between groups were analyzed by two-way ANOVA with the Dunnett's multiple comparisons test or one-way ANOVA with Bonferroni multiple-comparison test or paired Student's t tests. To test whether the changes of mRNA expression were rhythmic, one-way ANOVA followed by Bonferroni post hoc testing was performed to compare the average values for the peaks and troughs. We averaged the all data at 4 time points for each gene under each condition to approximate Midline Estimating Statistic Of Rhythm (MESOR) [20]. To test whether the experimental conditions significantly changed the overall mRNA expression levels, one-way ANOVA was used to compare the average values of four time points (08:00, 14:00, 20:00 and 02:00). All calculations were performed with Graph Pad Prism version 6.0 for Macintosh (MDF).

## Results

### Constant light exposure increases body weight without significantly increasing food intake

S1 Fig shows the body weights (A), liver weights (B), WAT weights (C) and food intakes (D) of the three groups of mice after 1, 2 and 3 weeks of treatment. At the end of the experiments (3 weeks), the LL mice with or without time-restricted feeding were 5.2% (p<0.001, n = 16) heavier than the LD mice (S1A Fig). All the mice survived the procedures and ostensibly healthy. The oscillations of liver and WAT weight were robust under the LD condition, but suppressed under the LL condition (n = 4). Time-restricted feeding restored the diurnal pattern (S1B and S1C Fig). Under the LD condition, the mice ate 5-fold more food at dark phase than at light phase (p < 0.001, n = 16) (S1D Fig) during the experiments. Under the LL condition without time-restricted feeding, peak feeding occurred at 07:00–19:00 at 1 and 3 weeks, while it occurred 19:00–07:00 at 2 weeks. However, the difference in food intake between the two phases was relatively small. Under the LL condition with time-restricted feeding, the mice ate only at 19:00–07:00 as expected. There was no difference in the total daily food intake among the three conditions.

### Effects of time-restricted feeding on lipid contents of the liver and WAT and plasma levels of lipids and glucose

Hepatic TG contents cycled robustly under the LD condition, peaking at 14:00, but the oscillation was suppressed under the LL condition (S2A Fig) (n = 4). Time-restricted feeding partially restored the diurnal pattern. No such rhythms were observed for hepatic TC and NEFA contents (S2B and S2C Fig). Time-restricted feeding decreased hepatic NEFA contents under LL cycle compared with other conditions, especially at 08:00. In contrast, no discernible rhythms were observed in the TG, NEFA and glycerol contents of the WAT (S2D, S2E and S2F Fig).

Plasma TG levels showed a diurnal rhythm reaching a trough at 20:00 under the LD condition (n = 4) (S2G Fig). The LL condition shifted the trough to an earlier point 14:00 and plasma TG levels were significantly higher than those under the LD condition at 20:00 and 02:00. Time-restricted feeding attenuated the high TG levels which was evoked by the LL condition at 20:00 and 02:00, but did not affect the overall rhythm. Plasma NEFA levels also showed a diurnal rhythm reaching a trough at 02:00 under the LD condition (S2H Fig). The LL condition shifted the trough to an earlier point, 20:00. Plasma NEFA levels under the LL condition were significantly lower than those under the LD condition at 08:00. Time-restricted

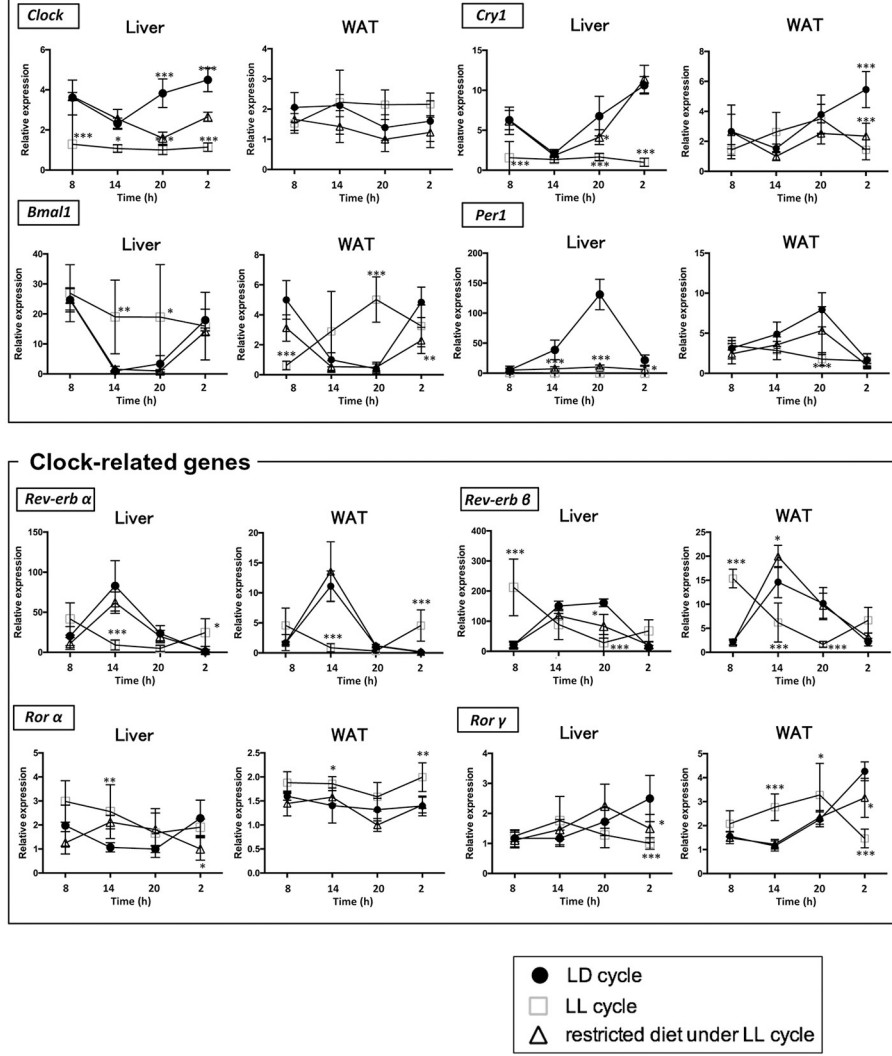

**Fig 1. Effects of constant light exposure with and without time-restricted feeding on the rhythmic mRNA expression of the representative clock and clock-related genes in liver and WAT.** Mice were housed under three conditions (closed circle, *ad libitum* under LD cycle; open square, *ad libitum* under LL cycle; open triangle, restricted feeding at night-time under LL cycle). The liver was isolated at 4 time points (08:00, 14:00, 20:00 and 02:00) for mRNA isolation. mRNA levels of the following genes (*Clock*, *Bmal1*, *Cry1*, *Per1*, *Rev-erbα*, *Rev-erbβ*, *Rorα* and *Rorγ*) were measured by qRT-PCR and expressed as values relative to the 08:00 time point, which is arbitrarily defined as 1, Values indicate the mean ± SD of 4 mice per group for each time point. Primer sequences are shown in Table 1. * p < 0.05, ** p < 0.01, *** p < 0.001, LD cycle vs LL cycle, LD cycle vs restricted feeding under LL cycle determined by two-way ANOVA followed by the Dunnett's multiple comparisons test.

feeding did not affect the diurnal rhythm. In contrast to the plasma levels of TG and NEFA, a diurnal rhythm was not observed for plasma levels of TC under any of the three conditions (S2I Fig). Time-restricted feeding raised plasma TC levels at all time points.

Plasma glucose levels showed a diurnal rhythm peaking at 14:00 under the LD condition. The LL condition shifted the peak to a later point, 02:00. Plasma glucose levels of the LL mice were significantly higher than those of the LD mice at 02:00 (S2J Fig). Time-restricted feeding fully restored the diurnal pattern.

## Constant light exposure eliminates circadian rhythms of the clock genes and lipid metabolism genes in the liver, while time-restricted feeding restores the disrupted circadian rhythms

To know whether disruption of the rhythms in the liver caused by constant light exposure is improved by restricted feeding at night-time in mice, we measured the mRNA expressions of clock genes (*Clock*, *Bmal1*, *Cry1 and Per1*) and clock-related genes (*Rev-erbα*, *Rev-erbβ*, *Rorα* and *Rorγ*) in the liver. Under the LD condition, clock genes and clock-related genes had robust diurnal expression patterns in the liver (p < 0.05, n = 4) (Fig 1). The LL condition significantly decreased the amplitudes of the rhythms of clock genes and clock-related genes. In addition, the LL condition profoundly decreased the average expression of *Clock*, *Cry* and, *Per1* in the liver (n = 16) (Fig 2). Conversely, time-restricted feeding restored the diurnal patterns of most

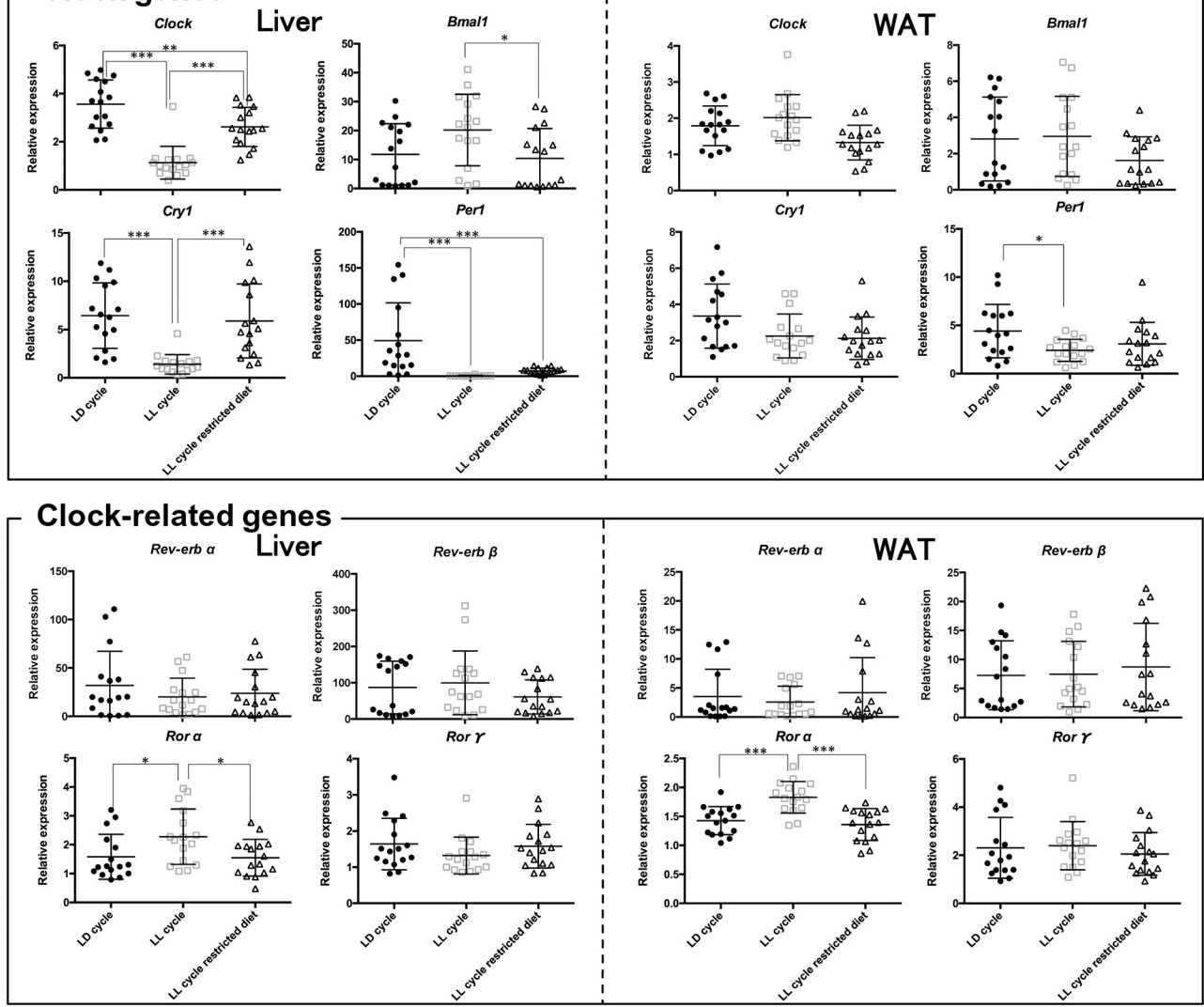

**Fig 2. Effects of constant light exposure with and without time-restricted feeding on the average mRNA expression levels of the clock and clock-related genes in liver and WAT.** The average values of four time points shown in Fig 1 were compared between three conditions: *ad libitum* under LD cycle, *ad libitum* under LL cycle, and restricted feeding at night-time under LL cycle. * p < 0.05, ** p < 0.01, *** p < 0.001, LD cycle vs LL cycle, LD cycle vs restricted feeding under LL cycle determined by one-way ANOVA with Bonferroni multiple-comparison test.

of the clock genes and clock-related genes except *Per1* even in the presence of light exposure (Fig 1).

Next, we measured the mRNA expressions of genes involved in lipid metabolism (cholesterol synthesis, fatty acid synthesis, triglyceride synthesis, neutral lipids hydrolysis, transcriptional factors) in the liver. Under the LD condition, all of these genes except *Lxrα* and *Srebp1c* had robust diurnal expression patterns in the liver (p < 0.05, n = 4) (Fig 3). The LL condition profoundly decreased the average expression of all of the genes except *Pparγ* (n = 16) (Fig 4). In addition, the LL condition significantly decreased the amplitudes of the oscillations of most of the genes involved in lipid metabolism. Conversely, time-restricted feeding restored the diurnal patterns of the genes to various degrees even in the presence of constant light exposure (Fig 3). Time-restricted feeding restored the average expression of all the genes except *Specificity protein 1* (*Sp1)* (Fig 4).

## Constant light exposure eliminates circadian rhythms of the clock genes and genes involved in lipid metabolism in WAT, while time-restricted feeding restores the disrupted circadian rhythms

We measured the mRNA expressions of the clock genes and clock-related genes in WAT. Under the LD condition, most of the clock genes and clock-related genes had robust diurnal expression patterns in WAT (p < 0.05, n = 4) (Fig 1). However, the differences between the peak and trough values were not significant for *Clock* or *Rorα* (p > 0.05, n = 4), indicating that the expression of the genes was not rhythmic. The LL condition significantly attenuated the diurnal patterns of all the genes in WAT, but its effects were much milder than in the liver. The LL condition significantly decreased the average expression of only *Per1*, but it did not decrease the expression of the other clock genes. The average expression of *Rorα* was increased (n = 16) (Fig 2).

Next, we measured the mRNA expressions of the genes involved in lipid metabolism in WAT. Under the LD condition, the expressions of these genes also had robust diurnal expression patterns in the WAT (p < 0.05, n = 4) (Fig 3). However, the differences between the peak and trough values were not significant for *Hmgcr*, *Atgl*, *Srebp1c*, *Srebp2* or *Sp1* (p > 0.05, n = 4), indicating that the expression of the genes was not rhythmic. The LL condition significantly attenuated the diurnal patterns of all the genes, but its effects were much milder than in the liver. In addition, LL condition did not decrease the average expression of any of the genes involved in lipid metabolism. The average expressions of *Dgat2 and Fasn* were increased instead (n = 16) (Fig 4). Conversely, time-restricted feeding restored the diurnal patterns of some of the genes involved in lipid metabolism (*Dgat2*, *Lipe*, *Pparα* and *Pparγ*) in WAT to various degrees even in the absence of constant light exposure (Fig 3). The effects were more modest than those in the liver.

## Discussion

In this study, we investigated the effects of constant light exposure and time-restricted feeding on the peripheral circadian rhythms of the clock genes and the genes involved in lipid metabolism in the liver and WAT of mice. A rhythmic expression was observed for most of the genes in the liver and to a lesser degree in the WAT under LD cycle. Circadian clock genes were coordinately expressed in these organs as reported previously [21]. Constant light exposure eliminated rhythmic expression of almost all the genes in both tissues. Interestingly, constant light exposure markedly decreased the average expression of *Clock*, *Cry*, *Per1* and most of the genes involved in lipid metabolism in the liver, whereas such effects were not observed in the WAT. Time-restricted feeding restored the rhythms to various degrees even in the presence of

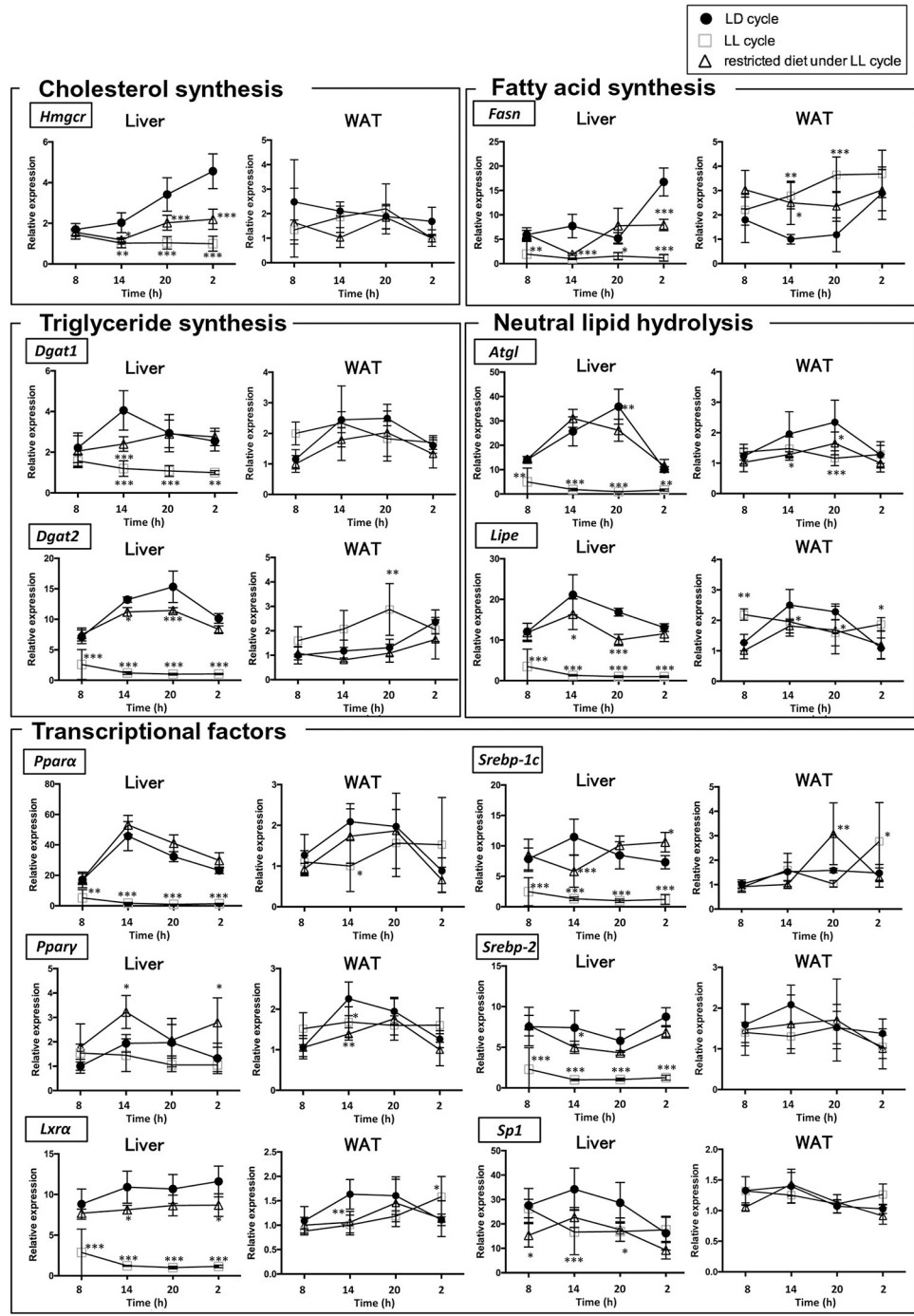

**Fig 3. Effects of constant light exposure with and without time-restricted feeding on the rhythmic mRNA expression of the genes involved in lipid metabolism in liver and WAT.** Mice were housed under three conditions (closed circle, *ad libitum* under LD cycle; open square, *ad libitum* under LL cycle; open triangle, restricted feeding at night-time under LL cycle). The WAT was isolated at 4 time points (08:00, 14:00, 20:00 and 02:00) for mRNA isolation. mRNA levels of the following genes (*Hmgcr*, *Dgat1*, *Dgat2*, *Atgl*, *Lipe*, *Fasn*, *Lxrα*, *Pparα*, *Pparγ*, *Srebp1c*, *Srebp2* and *Sp1*) were measured by qRT-PCR and expressed as values relative to the 08:00 time point, which is arbitrarily defined as 1, Values indicate the mean ± SD of 4 mice per group for each time point. Primer sequences were shown in Table 1. * $p < 0.05$, ** $p < 0.01$, *** $p < 0.001$, LD cycle vs LL cycle, LD cycle vs restricted feeding under LL cycle determined by two-way ANOVA followed by the Dunnett's multiple comparisons test.

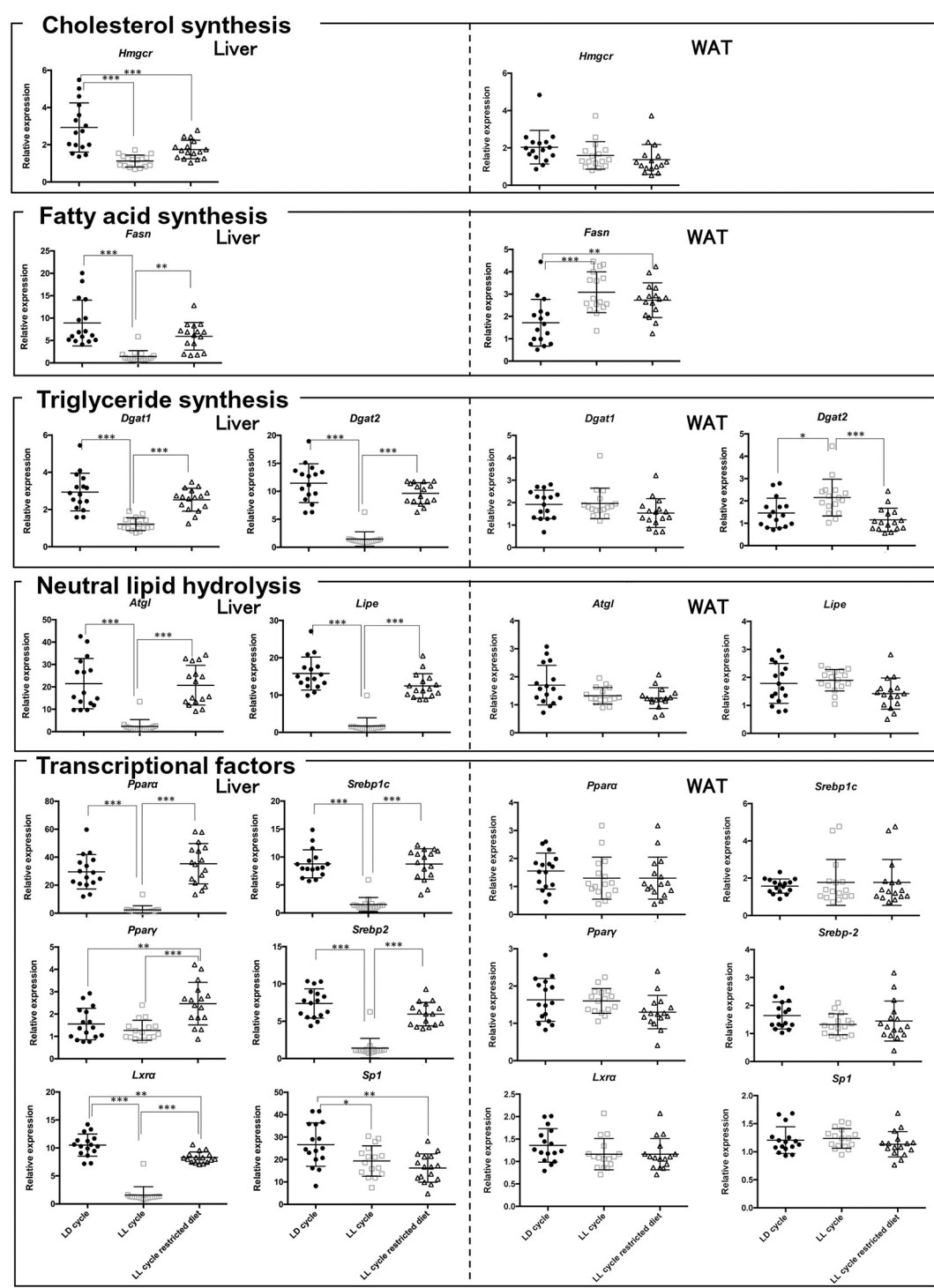

**Fig 4. Effects of constant light exposure with and without time-restricted feeding on the average mRNA expression levels of the genes involved in lipid metabolism in liver and WAT.** The average values of four time points shown in Fig 3 were compared between three conditions: *ad libitum* under LD cycle, *ad libitum* under LL cycle, and restricted feeding at night-time under LL cycle. * p < 0.05, ** p < 0.01, *** p < 0.001, LD cycle vs LL cycle, LD cycle vs restricted feeding under LL cycle determined by one-way ANOVA with Bonferroni multiple-comparison test.

constant light exposure, indicating that the effects of time-restricted feeding were dominant over the effects of constant light exposure for many genes involved in lipid metabolism in both liver and WAT.

Constant light exposure abrogated the rhythmic expression of *Bmal1*, *Cry1* and *Per1* in both the liver and WAT of the mice (Fig 1). Time-restricted feeding markedly restored the rhythmic expression of *Bmal1* and *Cry1*, but not *Per1* in the liver, while it restored the rhythmic expression of *Bmal1* and *Per1*, but not *Cry1* in the WAT. These results indicate that nutrient cues are dominant over the neural input from the SCN in determining the rhythmic expression of *Bmal1* and *Cry1* in the liver and *Bmal1* and *Per*1 in the WAT. However, the neural input from SCN may be dominant over nutrient cues for *Per1* in the liver and *Cry1* in the WAT. In this regard, Polidarova et al. reported that time-restricted feeding restored the rhythmic expression of *Per1* in the liver [22]. They exposed the mice to constant light for 30 days and time-restricted feeding for the last 14 days, whereas we exposed the mice to time-restricted feeding for 21 days under constant light exposure. The difference in the experimental condition may account for the different results. With regard to the mechanisms behind the food-dependent peripheral clock resetting, changes in metabolites such as glucose, NAD$^+$ and metabolic hormones such as ghrelin, insulin, glucagon and oxyntomodulin have been proposed as determinants of the food-dependent circadian rhythms [2, 23–26]. REV-ERBs and RORs activate and repress the expression of *Bmal1*, respectively. The rhythmic expression of *Rev-erbα* and *Rev-erbβ* were opposite to that of *Bmal1*, and the rhythmic expression of *Rorα* and *Rorγ* were synchronized with that of *Bmal1* (Fig 1). Constant light exposure abrogated the rhythmic expression of *Rev-erbα*, *Rev-erbβ*, *Rorα* and *Rorγ* in both the liver and WAT. Time-restricted feeding markedly restored the rhythmic expression of *Rev-erbα*, *Rev-erbβ* and *Rorγ*, but not *Rorα* in the liver. These results indicate that nutrient cues are dominant over the neural input from the SCN in determining the rhythmic expressions of *Rev-erbα*, *Rev-erbβ* and *Rorγ*.

Circadian rhythms of lipid metabolism are generated in part by the rhythmic gene expression of several metabolic enzymes [27]. The mRNA expression of *Pparα* which transactivates the genes primarily involved in fatty acid catabolism, has a robust rhythm under the LD conditions in both the liver and WAT as reported previously [28]. The LL condition abrogated this rhythm, but time-restricted feeding strongly restored the rhythmic expression even under constant light exposure (Fig 3). According to Oishi et al. [16], Clock/Bmal1 heterodimer transactivates the *Pparα* gene via an E-box-rich region, thereby contributing to the generation of rhythmic expression of *Pparα*. This was confirmed by the elimination of the rhythmic expression of hepatic *Pparα* in *Clock* mutant mice. Conversely, PPARα is required for the expression of *Bmal1*. Therefore, these two genes are under reciprocal control [29]. There was a lag time between the peaks for the oscillation of the two genes, probably reflecting the time required for accumulation of the transcripts of the genes. Interestingly, the oscillations of *Bmal1* and *Pparα* were completely abrogated by constant light exposure and restored by restricted feeding (Figs 1 and 3). These results imply that the expression of these genes is strongly influenced by the diurnal regulation of SCN. More importantly, the SCN-mediated abrogation of the normal oscillations can be totally reversed by nutritional cues. It is not surprising to note that *Dgat2* showed distinctive responses to both constant light exposure and restricted feeding (Fig 3), because *Dgat2* is negatively regulated by a PPARα ligand [30].

The oscillation of *Lipe* and *Atgl* was also completely abrogated by constant light exposure and completely restored by restricted feeding (Fig 3). Similar rhythmic expression of these genes in WAT has been reported [27]. Like *Pparα*, *Atgl* and *Lipe* are directly regulated by the Clock/Bmal1 heterodimer in an E-box-dependent manner in adipose tissue [27]. Although ATGL and LIPE significantly contribute to TG hydrolysis in WAT [11], it is noteworthy that their contribution to TG hydrolysis in the liver is relatively marginal [31, 32].

Although the expression levels of *Srebp1c* and *Srebp2* were significantly decreased by constant light exposure and restored by time-restricted feeding in the liver, we did not observe distinct rhythmic expression patterns for these transcription factors (Fig 3). Transcription of *Srebp*, which positively regulates the synthesis of cholesterol and fatty acids [14], is known to be controlled by *Clock* [33, 34]. The weakness of rhythms in the expression of *Srebp1c* and *Sreb*p2 is consistent with the expression pattern of *Clock* which does not oscillate in mice [35, 36].

Robust diurnal expression of *Hmgcr* and *Fasn* has been known for decades [18, 37]. These phenomena were recapitulated in our present study. Although constant light exposure eliminated the rhythmic expression of the genes, time-restricted feeding partially restored the rhythmic pattern in both tissues. *Hmgc*r and *Fasn* are positively regulated by *Srebp2* and *Srebp1c*, respectively. Although it is apparently inconsistent with the findings that both *Srebp2* and *Srebp1c* did not show robust rhythmic expression, the circadian rhythms of the genes can be explained by the circadian control of posttranslational proteolytic cleavage of SREBPs [33, 34]. Indeed, Insig2, which regulates proteolytic cleavage of *Srebp* by *Scap*, is transcriptionally regulated by circadian *Rorα/γ* [38]. The rhythmic expression patterns of *Rorα* and *Rorγ* were consistent with the expression patterns of *Hmgcr and Fasn* (Figs 1 and 3).

The most interesting finding of the present study is that constant light exposure markedly decreased the average expression of the genes in the liver, but not in the WAT (Figs 2 and 4). Instead, constant light exposure increased the average expression of *Dgat2* and *Fasn* in the WAT. To our knowledge, this phenomenon has not been previously reported. Previous studies which aimed to examine the effects of constant light exposure on the circadian rhythms did not directly examine the oscillation of hepatic genes [3, 4, 39]. In other models with disrupted circadian rhythms such as feeding a high fat diet (HFD) [7], *Bmal1*-deficient [40, 41] or *Per1/ Per2*-deficient mice [42], the mRNA expressions of some, but not all, of the genes were significantly decreased. The effects of constant light exposure appear unique in that they were much more generalized; it affected most of the genes. Generally, basic transcriptional activities of the house-keeping genes are determined by the Sp1 family proteins. In the present study, however, constant light exposure did not attenuate the basic transcription of *Sp1* as significantly as that of other genes (Fig 3). Since Sp1 is primarily modulated by post-translational modification [43], it is more likely that constant light exposure disrupts the basic transcriptional machinery by interfering with post-translational modification in a liver-specific manner. Further studies are warranted to unravel the mechanisms behind this intriguing phenomenon.

Hepatic TG content is regulated by various factors such as influx of plasma NEFA, lipogenesis, lipolysis and lipoprotein production. Although hepatic TG content oscillated rhythmically in the LD cycle with peak levels at 14:00 as reported previously [42], the rhythm was eliminated by constant light exposure (LL cycle) (S2A Fig). The elimination of oscillation in hepatic TG content was probably caused by the mice feeding at irregular times as a result of constant light exposure. It is uncertain which of the factors mentioned above is the major determinant of the hepatic TG content. Of course, the loss of rhythmic expressions of fatty acid metabolism-related genes such as *Dgat1*, *Dgat2*, *Fasn*, *Atgl*, *Lipe* and *Pparα* may contribute to the loss of rhythmic oscillation of hepatic TG contents. Although aberrant circadian rhythms are known to be associated with hepatic steatosis [44, 45], the hepatic TG contents of the mice under the LL condition were not as high as those observed in hepatic steatosis (S2A Fig).

With regard to plasma lipids, TG and NEFA levels showed a rhythmic oscillation under the LD condition. Constant light exposure changed the oscillation pattern (S2G and S2H Fig). These changes in the rhythms of plasma TG and NEFA were probably caused by arrhythmic feeding behavior, which are observed under the LL cycle [46]. The plasma levels of glucose and NEFA were higher during their light phase under the LD condition (S2H and S2J Fig), as was

reported in other rodent models [27]. Constant light exposure attenuated the diurnal rhythms, which may have resulted from the loss of rhythmic expression of the genes involved in fatty acid metabolism (Fig 3).

As reported previously [39], constant light exposure increased body weight primarily via increasing WAT mass (S1A Fig). The increase in body weight was not accompanied by a significant increase in food intake (S1D Fig), suggesting that the obesogenic effects of constant light exposure are largely due to a decrease in energy expenditure. These findings are consistent with the results of a previous study [39], although we did not directly measure energy expenditure or locomotor activity in the current study. It is noteworthy that time-restricted feeding did not reverse the increased body weight gain. Probably, time-restricted feeding could not override the LL-induced decrease in energy expenditure.

Our study has several limitations. First, the data of the rhythms of gene expression and organ weights were not collected individually, because the mice needed to be sacrificed for data collection. Second, 4 time points might be too few to detect some circadian rhythms of several parameters. Third, it is impossible to rule out that the time-restricted feeding affected the peripheral circadian rhythms via SCN or other pathways, because we did not examine circadian rhythms of SCN and other organs.

We were able to test our hypotheses in the experiments using as few as 16 mice in a group, implicating that our experimental design with minimal sacrifice can be used for further experiments on circadian rhythms to advance the principles of the replacement, refinement or reduction (the 3Rs) of the use of animals in research.

## Conclusions

Constant light exposure, in addition to eliminating the circadian rhythms of the clock genes, also eliminated the circadian rhythms in the genes involved in lipid metabolism in both liver and WAT. Constant light exposure also decreased the average expression of the genes in the liver, but not in the WAT, suggesting the presence of liver-specific transcriptional machinery which is sensitive to constant light exposure. Time-restricted feeding restores the circadian rhythms of most of the genes. Together, time-restricted feeding can be used to restore the circadian rhythms that are disrupted by the environmental stresses including constant light exposure which is afflicting modern human being.

## Supporting information

**S1 Fig. Effect of constant light exposure with and without time-restricted feeding on weight of body, liver and WAT and food intake.** (A) Changes in body weight (each group, n = 16). (B) liver weight and (C) WAT weight were measured at 3 weeks under different housing conditions (closed circle, *ad libitum* under LD cycle; open square, *ad libitum* under LL cycle; open triangle, restricted feeding at night-time under LL cycle). (D) Food intake was measured at 07:00–19:00 and 19:00–07:00 under different housing conditions. Values indicate the mean ± SD. * p < 0.05, ** p < 0.01, *** p < 0.001, LD cycle vs LL cycle, LD cycle vs restricted feeding under LL cycle determined by two-way ANOVA and one-way ANOVA test for A-C, and by paired Student's t tests and one-way ANOVA test for D.
(PDF)

**S2 Fig. Effects of constant light exposure with and without time-restricted feeding on the rhythms of plasma lipids and hepatic lipids.** (A) Triglyceride, (B) non-esterified fatty acids and (C) total cholesterol were measured in the liver housed under different conditions (closed circle, *ad libitum* under LD cycle; open square, *ad libitum* under LL cycle; open triangle,

restricted feeding at night-time under LL cycle) at each time points (08:00, 14:00, 20:00 and 02:00) using enzymatic colorimetric assay kits. Plasma was collected at each time points (08:00, 14:00, 20:00 and 02:00). (D) Triglyceride, (E) non-esterified fatty acids and (F) glycerol in WAT were measured by enzymatic colorimetric assay kits. Plasma (G) triglyceride, (H) non-esterified fatty acids, (I) total cholesterol and (J) glucose were measured. Values indicate the mean ± SD of n = 4 mice per group for each time point. $^{*}$ $p < 0.05$, $^{**}$ $p < 0.01$, $^{***}$ $p < 0.001$, LD cycle vs LL cycle, LD cycle vs restricted feeding under LL cycle determined by two-way ANOVA followed by the Dunnett's multiple comparisons test.
(PDF)

**S1 Checklist.**
(PDF)

## Author Contributions

**Conceptualization:** Shun Ishibashi.

**Data curation:** Daisuke Yamamuro.

**Formal analysis:** Daisuke Yamamuro.

**Funding acquisition:** Shun Ishibashi.

**Investigation:** Daisuke Yamamuro.

**Methodology:** Yusaku Iwasaki, Toshihiko Yada.

**Project administration:** Shun Ishibashi.

**Supervision:** Manabu Takahashi, Shuichi Nagashima, Shun Ishibashi.

**Visualization:** Daisuke Yamamuro.

**Writing – original draft:** Daisuke Yamamuro, Shun Ishibashi.

**Writing – review & editing:** Manabu Takahashi, Shuichi Nagashima, Tetsuji Wakabayashi, Hisataka Yamazaki, Akihito Takei, Shoko Takei, Kent Sakai, Ken Ebihara, Yusaku Iwasaki, Toshihiko Yada.

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
