## [Decision Letter · Decision Letter 0]

17 Mar 2020

PONE-D-20-02763

Constant light-induced attenuation of peripheral circadian rhythms and its reversal by time-restricted feeding in the liver and white adipose tissue of mice

PLOS ONE

Dear Dr Ishibashi,

Thank you for submitting your manuscript to PLOS ONE. After careful consideration, we feel that it has merit but does not fully meet PLOS ONE’s publication criteria as it currently stands. Therefore, we invite you to submit a revised version of the manuscript that addresses the points raised during the review process.  Both reviewers bring forth significant issues with regard to the experiments and their analysis which will need to be addressed for this manuscript to be considered for publication.

We would appreciate receiving your revised manuscript by May 01 2020 11:59PM. To enhance the reproducibility of your results, we recommend that if applicable you deposit your laboratory protocols in protocols.io, where a protocol can be assigned its own identifier (DOI) such that it can be cited independently in the future. For instructions see: http://journals.plos.org/plosone/s/submission-guidelines#loc-laboratory-protocols

We look forward to receiving your revised manuscript.

Kind regards,

Eric M Mintz, Ph.D.

Academic Editor

PLOS ONE

Journal Requirements:

2. As part of your revision, please complete and submit a copy of the ARRIVE Guidelines checklist, a document that aims to improve experimental reporting and reproducibility of animal studies for purposes of post-publication data analysis and reproducibility: https://www.nc3rs.org.uk/arrive-guidelines. Please include your completed checklist as a Supporting Information file. Note that if your paper is accepted for publication, this checklist will be published as part of your article.

Reviewers' comments:

Reviewer's Responses to Questions

**Comments to the Author**

1. Is the manuscript technically sound, and do the data support the conclusions?

Reviewer #1: Partly

Reviewer #2: Yes

2. Has the statistical analysis been performed appropriately and rigorously? 

Reviewer #1: Yes

Reviewer #2: No

3. Have the authors made all data underlying the findings in their manuscript fully available?

Reviewer #1: Yes

Reviewer #2: Yes

4. Is the manuscript presented in an intelligible fashion and written in standard English?

Reviewer #1: Yes

Reviewer #2: No

5. Review Comments to the Author

Reviewer #1: Yamamuro and coworkers study the interactive effects of light and food on the regulation of peripheral circadian clock gene rhythms in liver and adipose tissue. They further measure food intake and body weight regulation. For their comparison they combine constant light exposure (LL) and time-restricted feeding (RF) in mice. They show that LL dampens gene expression rhythms in liver and adipose, which is partially restored by additional RF. This is not reflected in body weight development, though group differences are overall subtle, and weight gain strongly interferes with developmental weight gain at this early age.

While I think this is an interesting topic of both biological and medical interest, my enthusiasm for the paper is strongly dampened by several technical issues that make the data difficult to interpret.

1. No control for SCN pacemaker function is provided for LL conditions. Were mice still rhythmic at the behavioral level? If yes, all gene expression data should be reported relative to activity onset. If not, can the authors exclude that they are merely observing desynchronization effects (within and between subjects) due to dysfunctional SCN output? How can results from different animals be compared under these conditions? How were ZT times determined without any zeitgeber reference under LL/ad lib conditions?

2. Considering that most of the studied genes are rhythmic at least under control conditions I do not think that simply averaging all data points over the whole circadian cycle is very meaningful. Not surprisingly, most of the comparisons in Fig. 5 yield no significant effects.

3. How can you determine amplitudes for genes expression of which was measured from different animals at each time point? You state these values were derived from data in Figs 3 & 4, but only one experiment is reported there. How did you come up with 4 amplitude values? Experiments in 3/4 would need to be repeated several times to determine independent amplitude values as presented in Fig. 6.

4. Gapdh was used as reference gene even though it was previously shown to be rhythmic under control conditions (Zhang et al. PNAS 2013 and others). Also, it is likely to be regulated by metabolic cues (such as restricted feeding) considering its role in glucose metabolism. Please provide validations with a different housekeeping gene – e.g. Actb or Eef1a.

5. Liver weight has previously been shown to be circadian (Sinturel et al. Cell 2017). This is likely true for adipose tissue, too. Please correct for potential circadian effects in your data.

6. Please use “cryptochrome”, not “cryptochrom”; “rev-erb” not “rev-erv”.

7. Provide irradiation and spectral composition data for LL conditions.

Reviewer #2: Constant light exposure disrupts both central and peripheral circadian rhythms; however, it is not known whether these effects are similar in the liver as in the white adipose tissue (WAT), or whether time-restricted feeding restores the circadian rhythms in WAT as previously reported in the liver. In the present manuscript the effects of constant light exposure and time-restricted feeding on the peripheral circadian rhythms of the clock genes and some genes involved in lipid metabolism were studied in the liver and WAT. Under LD most of the genes showed rhythmic diurnal expression in the liver and in the WAT, in both tissues these patterns were disrupted by constant light exposure in most of the studied the genes, and markedly decreased the overall expression of the genes in the liver (except for Bmal1), but not in the WAT. Most rhythmic patterns were restored by restricted feeding time under constant light exposure. According to the authors: “The most impressive finding of the present study is that constant light exposure markedly decreased the average expression of the genes in the liver, but not in the WAT. Constant light exposure rather increased the average expression of Dgat2 and Fasn in the WAT. To our knowledge, this phenomenon has not been reported in the previous literature.”

The study is relevant and in general well designed and conducted. The methods are sound and adequately address the question at hand. The conclusions are supported by the results. Nevertheless, there are several issues that need to be addressed before publication in PLOS ONE, as follows:

1) Besides the clock genes, it is not clear why the genes here studied were selected; are these all the genes involved in lipid metabolism? If not, why these and no other genes? Genes could be presented in categories such as clock genes, clock controlled genes, TG-related genes, TC-related, etc. A scheme or diagram presenting the relation among lipid metabolism, enzymes, genes of lipid metabolism and clock genes could be very useful.

2) Statistical analysis of rhythmicity is not adequate (t-test between peak and trough), it could be better an ANOVA followed by a post-hoc test, a cosinor analysis could also be applied. If there is some reason for not using any of these analysis it should be provided.

3) The figures are too complex to guide the reader to understand the main findings. I suggest redesigning them in order to illustrate the main findings of the study that lead to the conclusions of the manuscript. Remaining data could be shown as complementary information.

4) Previous issue affects also the description of the results, although the text is difficult to read is easier to follow than the figures. I suggest use the way genes’ results are summarized in the discussion (particularly those related to lipid metabolism) as a guide to present the results in the text and to present the figures.

5) In figures 1 and 2 there is no indication of which symbol represents which experimental group. Please verify that each symbol is used for the same group in all figures. In addition, select symbols that are easily recognizable even in very small panels.

6) The manuscript has some typos and grammatical issues that should be addressed. I suggest asking for professional proof reading of the manuscript before re-submitting for publication.

6. PLOS authors have the option to publish the peer review history of their article (what does this mean?). If published, this will include your full peer review and any attached files.

Reviewer #1: No

Reviewer #2: No

---

## [Author Response · Author response to Decision Letter 0]

30 Apr 2020

Responses to Reviewers

Reviewer #1: Yamamuro and coworkers study the interactive effects of light and food on the regulation of peripheral circadian clock gene rhythms in liver and adipose tissue. They further measure food intake and body weight regulation. For their comparison they combine constant light exposure (LL) and time-restricted feeding (RF) in mice. They show that LL dampens gene expression rhythms in liver and adipose, which is partially restored by additional RF. This is not reflected in body weight development, though group differences are overall subtle, and weight gain strongly interferes with developmental weight gain at this early age.

While I think this is an interesting topic of both biological and medical interest, my enthusiasm for the paper is strongly dampened by several technical issues that make the data difficult to interpret.

1a. No control for SCN pacemaker function is provided for LL conditions. 

→Response: We would agree with the reviewer that it is desirable to use locomotor activity of the mice under LL condition to estimate SCN pacemaker function. Unfortunately, however, we did not collect the behavioral data by an actogram. 

1b. Were mice still rhythmic at the behavioral level? If yes, all gene expression data should be reported relative to activity onset. If not, can the authors exclude that they are merely observing desynchronization effects (within and between subjects) due to dysfunctional SCN output? 

→Response: We cannot say that the mice were still rhythmic at the behavioral level, because we did not collect the behavioral data by an actogram. But we measured the amounts of food intake during 07:00-19:00 and 19:00-07:00, separately. As shown in old Fig. 1D (New S1 Fig. D), food intake behavior was severely disturbed. According to Lamont et al. (Lamont EW et.al., Eur. J. Neurosci., 39 (2): 207-217, 2014), the behavior was somehow rhythmic at least in the first 10 days after switching from LD to LL cycle with prolonged period. After 1 month, the behavior was no longer rhythmic. We collected the data at 3 weeks after switching from LD to LL. Therefore, it is suspected that the behavioral rhythmicity was almost disrupted at this time point. Therefore, the arrhythmic expression of the genes during LL might reflect the desynchronization of SCN output.

1c. How can results from different animals be compared under these conditions? 

→Response: As we explain below, we would express the results based on actual time, instead of ZT times.

1d. How were ZT times determined without any zeitgeber reference under LL/ad lib conditions?

→Response: Technically, it should be difficult to collect organs at the same ZT adjusted by the behavior even if the behavior is somehow rhythmic at an earlier time point. Based on these considerations, we have replaced “ZT” with actual time in the relevant parts. We also have added these considerations in the revised manuscript (New p. 4, lines 22~23). 

2. Considering that most of the studied genes are rhythmic at least under control conditions I do not think that simply averaging all data points over the whole circadian cycle is very meaningful. Not surprisingly, most of the comparisons in Fig. 5 yield no significant effects.

→Response: MESOR (Midline Estimating Statistic Of Rhythm) would probably be better than an average if we had more data points. With as few as 4 data points, we were unable to determine the peaks and troughs. In addition, we were able to obtain only single value for MESOR by subtracting the mean value of troughs from the mean value of peaks, making statistical analyses impossible. Therefore, we decided to average the values for all the points as a surrogate of MESOR. Please note that the graphs of MESOR (Fig. 1 for comment 2 of reviewer #1 in Figures for Reviewers) are very similar to the graphs for the average (New Fig. 2 and 4 in the revised manuscript). 

We believe that the comparisons in old Fig. 5 (New Figs. 2 and 4) are significant: under the LL condition, the expressions of 10 of the 12 liver genes were significantly suppressed, while none of WAT genes were suppressed. This is an important observation that has not been reported previously.

In the revised manuscript, we added a brief rationale for using the average instead of MESOR (New p. 6, lines 10-12).

 3. How can you determine amplitudes for genes expression of which was measured from different animals at each time point? You state these values were derived from data in Figs 3 & 4, but only one experiment is reported there. How did you come up with 4 amplitude values? Experiments in 3/4 would need to be repeated several times to determine independent amplitude values as presented in Fig. 6.

→Response: We agree that old Fig. 4 was pointless and deleted the figure.

4. Gapdh was used as reference gene even though it was previously shown to be rhythmic under control conditions (Zhang et al. PNAS 2013 and others). Also, it is likely to be regulated by metabolic cues (such as restricted feeding) considering its role in glucose metabolism. Please provide validations with a different housekeeping gene – e.g. Actb or Eef1a.

→Response: Actually, Gapdh appeared to be a better choice than Actb (Nakao R, et al. Mol Genet Metab. 2017;121:190-7). We added the following to Methods:

“Mouse glyceraldehyde 3-phosphate dehydrogenase (Gapdh) mRNA was used as the invariant control. Although it was reported that Gapdh exhibited circadian oscillation in both liver and WAT, the oscillations were not statistically significant at least when the mice were fed during nighttime and appeared less robust than those of many other genes including Actb. Indeed, we did not find significant oscillation of Gapdh in both liver and WAT (data not shown).”

Actually, we reanalyzed the data corrected by Actb (Fig. 2 for comment 4 of reviewer #1 in Figures for Reviewers). Please find that the graphs corrected for Actb looked indistinguishable from those corrected for Gapdh (New Fig. 1 and 2). Moreover, we could not find a significant rhythms in the Ct values of Gapdh in the liver and WAT (Fig. 3 for comment 4 of reviewer #1 in Figures for Reviewers).

We could not find the suggested paper (Zhang et al. PNAS 2013 and others).

5. Liver weight has previously been shown to be circadian (Sinturel et al. Cell 2017). This is likely true for adipose tissue, too. Please correct for potential circadian effects in your data.

→Response: As pointed out by the reviewer and reported previously (Sinturel F. et.al., Cell, 169, 651–663, 2017), we observed circadian rhythms in the weight of the liver and WAT. These circadian rhythms of tissue weight under LD conditions disappeared under LL conditions and restored by time-restricted feeding. (New S1 Fig. B and C; Fig. 4 for comment 5 of reviewer #1 in Figures for Reviewers). Please find the new figures in the revised manuscript. 

We had already corrected the lipid contents in the organs by the weights of organs at each time point (New S2 Fig. A~F). 

6. Please use “cryptochrome”, not “cryptochrom”; “rev-erb” not “rev-erv”.

→Response: Done.

7. Provide irradiation and spectral composition data for LL conditions.

→Response: We added the following sentences to the methods: "Light conditions in the breeding room were as follows: Two 32W fluorescent lamps FHF32EX-N-H (Panasonic, Japan) were attached to the ceiling at two places in a 6.5 feet × 20 feet room. Light irradiation was 3520 lumens. The ceiling lamps were about 3.3 feet above the cages." (New p. 4, lines 19-22).

 

Reviewer #2: Constant light exposure disrupts both central and peripheral circadian rhythms; however, it is not known whether these effects are similar in the liver as in the white adipose tissue (WAT), or whether time-restricted feeding restores the circadian rhythms in WAT as previously reported in the liver. In the present manuscript the effects of constant light exposure and time-restricted feeding on the peripheral circadian rhythms of the clock genes and some genes involved in lipid metabolism were studied in the liver and WAT. Under LD most of the genes showed rhythmic diurnal expression in the liver and in the WAT, in both tissues these patterns were disrupted by constant light exposure in most of the studied the genes, and markedly decreased the overall expression of the genes in the liver (except for Bmal1), but not in the WAT. Most rhythmic patterns were restored by restricted feeding time under constant light exposure. According to the authors: “The most impressive finding of the present study is that constant light exposure markedly decreased the average expression of the genes in the liver, but not in the WAT. Constant light exposure rather increased the average expression of Dgat2 and Fasn in the WAT. To our knowledge, this phenomenon has not been reported in the previous literature.”

The study is relevant and in general well designed and conducted. The methods are sound and adequately address the question at hand. The conclusions are supported by the results. Nevertheless, there are several issues that need to be addressed before publication in PLOS ONE, as follows:

1) Besides the clock genes, it is not clear why the genes here studied were selected; are these all the genes involved in lipid metabolism? If not, why these and no other genes? Genes could be presented in categories such as clock genes, clock controlled genes, TG-related genes, TC-related, etc. A scheme or diagram presenting the relation among lipid metabolism, enzymes, genes of lipid metabolism and clock genes could be very useful.

→Response: One of the aims of this study is to clarify the effects of environmental cues (light/dark cycle and time-restricted feeding) on the development of non-alcoholic fatty liver disease (NAFLD). Therefore, we selected the genes which are potentially involved in the pathophysiology of NAFLD. In the liver, genes involved in de novo lipogenesis (DNL) such as SREBP1c, LXR and lipogenic genes, hydrolases of neutral lipids, and beta oxidation of fatty acids are important. In the WAT, genes involved in lipolysis such as ATGL and HSL are important. We have rewritten the introduction to make this aim clearer to the readers (New p. 3, 3rd paragraph and new p. 4, 1st paragraph). 

To further clarify, in the results and figures, we divided the genes into six categories: clock genes, clock-related genes, fatty acid synthesis genes, cholesterol synthesis genes, triglyceride synthesis genes, neutral lipids hydrolysis genes and transcriptional factors.

2) Statistical analysis of rhythmicity is not adequate (t-test between peak and trough), it could be better an ANOVA followed by a post-hoc test, a cosinor analysis could also be applied. If there is some reason for not using any of these analysis it should be provided.

→Response: As the reviewer suggested, cosinor analysis may be most appropriate for the analyses of the rhythmic phenomena in general. However, we collected data at only 4 points from different individuals because we needed to sacrifice the animals to collect the organs, and this was not sufficient for a cosinor analysis. Therefore, we replaced the t-test with ANOVA followed by a post-hoc test (New p.6, 2nd paragraph). 

The results were essentially similar to the previous results.

3) The figures are too complex to guide the reader to understand the main findings. I suggest redesigning them in order to illustrate the main findings of the study that lead to the conclusions of the manuscript. Remaining data could be shown as complementary information.

→Response: We agree that the data were not presented well in the figures. We moved old Figs. 1 and 2 to Supporting information (New S1 Fig and S2 Fig) and deleted the original Fig. 6.” Now the main results are shown in Figs. 1-4.

4) Previous issue affects also the description of the results, although the text is difficult to read is easier to follow than the figures. I suggest use the way genes’ results are summarized in the discussion (particularly those related to lipid metabolism) as a guide to present the results in the text and to present the figures.

→Response: As suggested, in the results and figures, we grouped the genes into the six categories mentioned above.

5) In figures 1 and 2 there is no indication of which symbol represents which experimental group. Please verify that each symbol is used for the same group in all figures. In addition, select symbols that are easily recognizable even in very small panels.

→Response: This was fixed. "

6) The manuscript has some typos and grammatical issues that should be addressed. I suggest asking for professional proof reading of the manuscript before re-submitting for publication.

→Response: After correcting typos and grammatical errors, we have asked a native English-speaking editor to brush up the English again before re-submission.

---

## [Editor Report · Decision Letter 1]

27 May 2020

Peripheral circadian rhythms in the liver and white adipose tissue of mice are attenuated by constant light and restored by time-restricted feeding

PONE-D-20-02763R1

Dear Dr. Ishibashi,

We are pleased to inform you that your manuscript has been judged scientifically suitable for publication and will be formally accepted for publication once it complies with all outstanding technical requirements.

With kind regards,

Hervé Guillou

Academic Editor

PLOS ONE
---

## [Editor Report · Acceptance letter]

1 Jun 2020

PONE-D-20-02763R1 

Peripheral circadian rhythms in the liver and white adipose tissue of mice are attenuated by constant light and restored by time-restricted feeding 

Dear Dr. Ishibashi:

I am pleased to inform you that your manuscript has been deemed suitable for publication in PLOS ONE. Congratulations! Your manuscript is now with our production department. 

With kind regards,

on behalf of

Dr. Hervé Guillou 

Academic Editor

PLOS ONE